# Global Assessment of Bridge Passage in Relation to Oversized and Excessive Transport: Case Study Intended for Slovakia

**Jozef Gnap** [1,*] , **Juraj Jagelčák** [1], **Peter Marienka** [1], **Marcel Frančák** [1] **and Mária Vojteková** [2]

1 Department of Road and Urban Transport, Faculty of Operation and Economics of Transport and Communications, University of Zilina, SK-010 26 Zilina, Slovakia; juraj.jagelcak@fpedas.uniza.sk (J.J.); peter.marienka@fpedas.uniza.sk (P.M.); marcel.francak@fpedas.uniza.sk (M.F.)

2 Department of Quantitative Methods and Economic Informatics, Faculty of Operation and Economics of Transport and Communications, University of Zilina, SK-010 26 Zilina, Slovakia; maria.vojtekova@fpedas.uniza.sk

\* Correspondence: jozef.gnap@fpedas.uniza.sk; Tel.: +421-41-513-3500

**Abstract:** The development of an economy and, in particular, the construction of new infrastructure as well as industrial enterprises creates demand for the road transport of oversized freight that exceeds the maximum permissible total mass of vehicle combinations with its share on the axles. Failure to comply with the defined technological processes and a deficiency in the assessment of permitting such forms of transportation can have a large adverse effect, predominantly on the lifetime of bridges in a road network, which can have international implications as well. There is no legislation adopted by the EU Member States, which would at least partially unify the authorisation procedures of these forms of transportation and, therefore, it results in problems when crossing borders and leads to differences related to the assessment of bridge passages. If there is no systematic inspection of this kind of transportation, it can lead to permanent damage of these bridges as well. Currently, and not only in Slovakia but also in other states, the assessment of bridge passage for certain routes is used for heavy and oversized transportation. It means that if we use 100 transports, 100 assessments of individual routes are needed, although some are the same routes or the same vehicles/vehicle combinations used for a number of transports. Thus, the authors designed a global assessment for bridge passage in relation to heavy and oversized road transport while verifying it in the conditions of the EU Member State from Central Europe–Slovakia. Roads are full of different types of vehicles/vehicle combinations for which the axle loads and distances of the axles (wheelbases) are important. Thus, there were vehicle/vehicle combinations parameters (big data) observed, for which the routes relating to heavy and/or oversized transportation were assessed from 1 January 2016 to 31 December 2020 in Slovakia. The global assessment of bridge passage introduces an entirely new approach within the procedure for obtaining a special permission for road use as well as within transport use itself. Given the low presence of freight with an abnormal axle load or enormous total mass, it is appropriate to define the limited conditions under which it would be possible to implement the global assessment in practice as well.

**Keywords:** oversized transport; excessive transport; big data; assessment of bridge passage; cumulative axle load

## 1. Introduction and Literature Review

Transportation as a sector of the country's national economy aims to satisfy the shipment requirements of society. Development of the economy brings about a rise of different types of transport. Depending on the type of transport, and mainly the cargo, requirements for its speed, quality, and price, as well as financial support and staffing, are changed. One of the specific categories is the transport of oversized and/or super loads and/or heavy and oversized transport. However, this transport, depending on its

character, requires a higher amount of attention paid to the safety of all the transit modes when compared to "standard transport." [1].

Concerning mass, the bridges are limiting engineering constructions in road transport. They are negatively affected by several factors. Besides being affected by the freight itself, they can withstand corrosion and degradation, which also depends on the bridge's geographic position [2]. It must be said that there are a number of bridges—not only in Slovakia—which are at, or even above, the limit of their planned lifetime, and it can have an adverse effect on their load capacities.

When drafting new routes, or reconstructing old ones, it is also necessary to take into consideration, where relevant, the presence of heavy and oversized transport. Amending the already established infrastructure costs time and money and may even be impossible. Thus, it is necessary to define suitable requirements for transport infrastructure, especially for routes used for heavy and oversized transport [3]. Unfortunately, road infrastructure designers often do not have enough information about the parameters needed for trouble-free passage of heavy and oversized transport [4].

As also mentioned in the studies of Maurer and coll [5] and Rymsza and coll [6], the EU Member States turn to procedures according to standard EN 1991-2 when designing new bridges, or when considering their reconstructions. Depending on the drafting standards in each Member State, it will lead to an amendment of the former loading schemes as well as the results of the load capacity for bridges. Using vehicles with a total mass of 60 t on a given road is also analysed by a designed vehicle with the same mass.

If the bridges and roads are designed improperly, their maintenance costs in relation to heavy goods vehicles are high. In the case of illegal or uninspected operation of vehicles/vehicle combinations exceeding the maximum permitted axle loads, these costs increase exponentially. Based on the studies of Marwan and coll [7] and Jacob and coll [8], we can argue that problems with vehicle overloading are not only seen in Slovakia or Central Europe, but this issue is worldwide as well. One of the possible solutions for limiting the operation of overloaded vehicles/vehicle combinations may be a more efficient system of control, which can be achieved by dynamic weighing systems.

Godavarthy et al., in their article [9] from 2015, focus on roundabout passage in relation to special transport, mainly oversized/overweight vehicle combinations. They designed via Thorus software six standard roundabouts through which—by simulation via AutoTurn software—the oversized vehicles passed in order to observe the spatial requirements of the roundabout passage. This study assessed the efficiency of different solutions for oversized vehicles. There were four strategies of passage assessed: a typical roundabout and three alternative strategies for a roundabout's building and passage—opposite direction travel (ODT), fully traversable central island (FTCI), and straight passage through the central island. The results of this study show, when using the ODT and FTCI method, a reduction in needed total apron, which would have to be covered by the roundabout in relation to oversized vehicles—without these designs taken into consideration. However, this can only be achieved by unique arrangements, such as installation of a splitter island that is traversable and suitable for tires of trucks, removable signage that enables vehicles to use any traffic lane (same direction traffic and opposite traffic), or entering and exiting splitter islands, which enables oversized vehicles to use any traffic lanes to exit or enter a splitter, if needed; the vehicles shall turn left in the opposite direction of normal traffic without circulating the central island. Such travels are conditioned by cooperation with escort vehicles. The disadvantage of these designs is that it is impossible to plant verdure.

The transport of large and heavy mechanical products was observed by Petru et al. [10]. The article studied this kind of transport via GPS, according to which the authors created relevant models. The research results provide a possibility for verifying all the critical route points, such as bridges and roundabouts.

Meng et al.'s [11] research focused on the passage of oversized vehicles through roundabouts. The authors used Dijkstra's algorithm, which takes into consideration the

size of a turn angle. Based on the measurements, the authors designed the routes with minimum travel time.

Petraska et al. [12] analysed the problem of the optimisation of transport processes in relation to oversized and heavyweight loads not on roads but in rail transport. The authors designed a universal multi-criteria system of selection and assessment of rational routing for oversized and heavyweight loads carried by rails. Based on these principles, they designed a criteria methodology that enables an objective assessment of route segments of heavyweight and oversized loads transport process route segments for rail transport.

Critical points of heavy and oversized transport are level and elevated intersections, bridges, toll gates, traffic signage, and trolley and power lines, as well as other engineering structures in road infrastructure. Petru et al. [13], in their article, paid attention to the assessment of the transport routes of oversized and excessive loads in relation to the passage through roundabouts via simulation methods for vehicle movements.

Wolnowska and Konicki [14] also dealt with heavy and oversized load transports, aiming to evaluate the routes designed for Szczecin in Poland via the AHP method. They compared three simulated transports of steel construction by the analytic hierarchy process. The scientific contribution of their article lies in the rationalisation of urban transport of oversized and overweight loads in compliance with the concept of green areas.

The criteria system described by Bazaras et al. [15] is suitable for planning and designing the routes for heavy and oversized loads transport. This system enables one to objectively choose the most appropriate section of a route in the existing road network. It is also a tool for comparing different routes at a certain area, enabling one to choose the most suitable one according to their particular criteria via mathematical calculations. The article indicates that the criteria system is effective and may be used for the comparison of existing routes for oversized/overweight loads of different types of transport, with alternatives to build new sections of a route or to reconstruct the old ones.

Autelitano et al., in their article [16], pay attention to the transport of wind turbines and their parts in relation to heavy and oversized transport. They found out, via analysis, that road infrastructures have had to comply with a wide range of complex vehicle configurations for transporting wind turbine components that are often considered excessive and oversized by transport authorities. The result of the authors' activities and analyses is an overview of problems that occurred during the road transport of wind turbine components. Their suggestions include a strategy for operational planning based on the maximum width needed for a vehicle combination to travel with a turbine's rotor vanes in order to make the route identification easier. This methodology might be used by transport control authorities to promptly determine the possible unusual corridors for road transport.

Leclercq et al. [17] introduced a new concept of traffic operation management in cities on the principle of network division into several areas. The concept transferred information gained into a public map, being further interpreted for drivers to take a detour. The resulting solution provides a practical and valuable system for the reduction of urban traffic congestion.

The selection of appropriate routes for special transport was observed by Radomir [18], who applied a procedure in which the route had the lowest number of requests possible and the road and bridge load bearing capacity was not exceeded. These model situations considered vehicles with a width and height up to 7 to 8 m and with a mass up to 600 t.

Concerning the traffic situation in Poland, Mydlarz, and Wieruszewski, a study has been composed on the possibility of increasing the maximum permitted mass of combination vehicles when transporting wood material in Nordic countries. As well, the study took into account the effect of this solution on the economics of transport and ecology. This also relates to the study of Trzcinski and coll., which, besides this, concludes that the load capacity of vehicle combinations transporting wood material significantly fluctuates during the year, and thus, it is impossible, without knowing the issue in detail, to clearly determine how much cargo can be loaded without exceeding the maximum permitted mass [19,20].

To make oversized transport successful, it is necessary to ensure the use of special equipment and trained people, as well as the application of processes that are diametrically different from common road transport. Macioszek [21], in his article, defines the conditions for oversized load transport, characterizes the basic types of heavy and oversized transport with required papers, and includes a description of the preparation and organisation of such transports. His following article [22] describes the problems relating to the permits and fines used for oversized load transport, and the problems of transport security relating to the fitting techniques for oversized loads in road transport in Poland.

Nowadays, as mentioned in the study of Matuszkova and coll., it is crucial to reduce the costs of transport, given that exceeding the maximum permitted masses of vehicles/vehicle combinations can lead to the excessive damaging of the road network. Other factors to consider include the risk of exceeding the maximum permitted masses, which can increase the risk of a technical problem in the vehicle or a hazardous change in its driving performance [23].

Under Slovak conditions, the load capacity of bridges is determined by the procedures defined in Technical Conditions 104—Traffic loads on road bridges and footbridges. Within these requirements, there are five loading schemes established:

- normal load
- single axle load
- exclusive load
- exceptional load
- pedestrian load on footways or cycle tracks

Normal load is set as the maximum permitted mass of a single vehicle at normal load without a limited number and location of vehicles on the bridge. Currently, to determine the value of normal load, a quantity known as the "mass of the representative vehicle", which is given in 320 kN, corresponding to 32 t—i.e., the maximum permitted mass of a four-axle vehicle. There is also a quantity, known as the "factor of normal load", which is used, bearing in mind the ability of bridge construction to transmit the load caused by load model LM1 according to EN 1991-2.

Single axle load is set justifiably according to the decision of the designer's project or substitution project, or the administrator of an engineering construction. For single axle load, a load model 2 (LM2) is used according to article 4.3.3 of EN 1991-2.

Exclusive load is set as a maximum permitted mass of a single vehicle on the bridge. A vehicle model for exclusive load corresponds to the class of special vehicles 900/150 according to EN 1991-2, article A.2 part (1). The notation 900/150 of a special vehicle means that the total weight of this vehicle is 900 kN, composing of 6 axle-lines of 150 kN with partial 1.5 m wheelbases.

Exceptional load in Slovakia is set as an instantaneous maximum weight of a special vehicle transporting abnormal cargo that can be carried by bridges under the defined conditions for bridge passage. In EN 1991-2/NA1, these conditions are specified for travelling at a speed of 5 km/h, fully excluding other transport in optimum trace with a deviation of ±0.30 m. As a recommended load model for exceptional load, a vehicle with notation 3000/240 composed of 12 axle-lines of 240 kN and one axle-line of 120 kN is used, and the distance between axles is 1.5 m. The width of a load model is 4.5 m.

Abroad, there are different load models set for exceptional load. In Austria, there is a model of notation 3000/200 used, i.e., 15 axle-lines of 200 kN with total vehicle weight of 3000 kN, and, in the Czech Republic, there are three different load models used. Concerning highways, motorways, and special roads, load models with notation 3000/240 and travelling at a maximum speed of 5 km/h, excluding other kinds of transport, as well as 1800/20 travelling at a maximum speed of 70 km/h, excluding other kinds of transport, with a weight exceeding 5 kN are permitted. Again, concerning the 1st and 2nd class roads, there is a load model of notation 1800/200 with a maximum speed of 70 km/h used, fully excluding other kinds of transport. Load model 900/150 with a maximum speed of 70 km/h is used, while excluding other kinds of transport.

Another scientific paper from Petru et al. [24] focuses on the transport of oversized cargo from the urban road network's point of view. The authors analyse the transport of oversized cargo from the perspective of sustainable transport infrastructure in cities. The research results serve as a source for technical conditions that include procedures and technical recommendations in the Czech Republic, bearing in mind a sustainable, safe, and economical transport infrastructure. The authors analysed the results for the determination of parameters to ensure the passage of oversized cargo on roads. Video analysis, GPS devices, drones, modelling, and simulations were used for the analyses of the vehicle routes. This research deals with the issue of the transport of excessive and oversized cargo from the point of view of determining the parameters for roads. The authors also proposed technical recommendations for roundabouts passing, such as modifying the roundabout in the opposite direction, relocating the location of the poles, and increasing the rigidity of the structure of the dividing islands. This research was mainly focused on transport in cities; therefore, no analyses of bridge passing were proposed.

Zhang et al. [25], in their article, focus on the analysis and discussion of the values in relation to the dimensions, axle loads, and masses intended for road vehicles. The authors debate via comparative analysis which standard would be the most suitable for the real needs of road transport, as well as the requirements of road equipment within the country and abroad. They compare five totally different countries/continents with different legislations. These are China, Japan, Europe, the United States, and Australia. The comparison includes the width of the traffic lanes and the maximum permitted width of vehicle combinations according to their composition and maximum permitted heights. The comparison of the maximum permitted masses included the axle load at a given number of axles and the total weight. The results comprise views and proposals that will be useful in the revision of the standard GB 1589-2004 Limits of dimensions, axle load, and masses for road vehicles. Their proposal is based on the combination of the standard and real road situations in China, covering some features from other, developed countries.

Using different types of semi-trailers for heavy load transport enables transport companies to increase competitiveness. Figlus and Kuczynski [26], in their article, present the results from the analyses of semi-trailer operational damage, debating on the causation of damages and faults in relation to the haulage of long oversized loads. The study uses a multi-criteria methodology of decision-making to select a semi-trailer for excessive load transports. The results conclude that the most suitable solution is a selection of expandable semi-trailers for long oversized loads.

Similarly, Corbally et al. [27], in their article, focus in detail on the issue in Ireland relating to oversized and excessive transport primarily across bridges. They analysed the legislation in each foreign state. It was followed by an in-depth analysis of this issue as it relates to bridge passage by measuring the effects that occurred as well. The results of their study enable one to make informed decisions about the issue around permission and the exclusive use of roads. This should enhance and unify the current procedures for issuing permits.

Vehicle overloading has a significant impact on road infrastructure. Agbelie et al. studied this issue and proposed a methodology consisting of a technique that correlates the designed AASHTO vehicles with FHWA vehicle classes, estimates the marginal life cycle and usage costs of bridges, and assigns these costs to each vehicle class according to the configuration of axles and kilometres driven. The research results can help road infrastructure administrators to create a policy for transport authorisation in terms of bridge damage due to overweight vehicle use [28].

Petraska et al., under Lithuanian conditions, proposed a methodology of selection for the heavy and oversized freight transport system through minimum financing provided. The resulting solution took in mind the use of multimodal transport for this freight [29].

International oversized transport may be demanding in terms of legislation. Badescu et al. paid attention to this issue, and they identified the characteristics of different legislation within the scope of international oversized transport. They aimed to design

and implement less heavy and faster oversized and excessive transport with a lower environmental footprint and reduced human and material resources [30].

Zsamboky describes the advantages and challenges occurring in oversized and excessive transport in the USA [31]. He points out the costs, delivery time, and quality relating to the transport, and highlights that even oversized load transport is more money-consuming and technologically demanding. However, it is also more favourable than shipments divided into smaller parts.

Ryczynski et al.'s [32] conference article was dedicated to the issue of the suitable selection of a vehicle for oversized and excessive load transport. They mostly described the problems within the transport of oversized loads in the army and, thus, identified the most acute risks arising from this kind of transport. Based on this, there was a conception model for risk assessment proposed.

Safety is a key factor in oversized transport. Palaitis et al. in their article, when assessing the risk quantitatively and economically, used a probability theory and mathematical statistics were used, in the case of ambiguous options, for describing the situations or processes [33].

Pashkevich et al.'s [34] article was devoted to the peculiarities of delivery of large and heavy goods by road transport. The research collective introduced a concept of decision support system together with the interaction of its modules. Based on the algorithm, the software complex can select a suitable vehicle and foresee the consequences of the solutions suggested.

Hanzl within the Czech Republic identified the critical places that occurred in the transport of oversized cargo [35]. These places have a negative impact on vehicle and traffic safety. Based on these findings, the author summarized several measures to increase road infrastructure and traffic safety.

Kokkalis et al. [36], in their article, observed the effect of the transport of oversized/overweight vehicles along the motorway in Greece, as well as its related road infrastructure conditions. The authors studied the procedure of obtaining the permits for excessive load delivery and proposed to amend the permit fee policy, aiming to cover the entire financial burden for this road transport.

Zong et al. [37] established a method of safety evaluation for overweight/oversized cargo transport with the use of advanced technology incorporating a sensor and analytic hierarchy process. The authors, during the measurements, gained information on the road gradient and width, height of obstacles, and weight.

Paulauskas et al. [38] provided an alternative to road or rail oversized load transport in the form of inland waterway shipping. The research collective analysed the potential and the feasibility of inland waterway shipping of oversized cargo and designed a new adjustment for the infrastructure intended for this kind of transport.

Melnyk et al. [39] proposed a methodology for the selection of vessels, which considered their suitability for oversized cargo transport. The authors considered the net present value and profitability index for the effectiveness assessment of specialized vessel acquisition and operation projects intended for heavy-weight load transport.

Onyshchenko et al. [40], in their study, identified a system of potential negative impact factors on the ship's operational condition during transport of oversized and heavy cargo. The authors divided the negative factors into two categories—factors occurring during loading and unloading and factors occurring during carriage.

Prodon et al. [41] describes the process of the transport of large pre-built accelerator components and physics detectors, which are considered oversized cargo. During the transport, a maximum acceleration of 0.1 g and maximum tilt of 1 degree was allowed. This was achieved with the use of transport monitoring. Multiple configurations of vehicle combinations were used, while the biggest trailer configuration had a length of 64 m and a weight of 448 t. Huang and Han [42] developed a model based on an entropy weight method, cloud model, and TOPSIS method for the selection of an optimal urban transport route for oversized transport from among alternative transport routes with more accuracy

and objectivity. Their model considers the subjective and objective weights. This research can be used for the selection of large cargo transportation routes in urban territories, but it only considers oversized transport (e.g., clearance under the bridges) based on the present results from four routes.

Petraska et al. [4] developed an algorithm for the assessment of heavyweight and oversized cargo transportation routes, which may be considered as a new science approach. Literature research revealed that there is no universal criteria system for route selection for oversized/excessive cargo transport. Their research and analysis allowed them to develop a new system of criteria that was essential for allowing objective assessment of cargo transport processes and comparing the different modes of transport, route sections, and transport and transhipment technologies, and may be adapted to virtually any territory. This new criteria system is not only appropriate for assessing existing cargo transport possibilities in the territory but is also appropriate for planning the long-term routes of transportation of such cargo pursuant to the economic development promotion criteria. This research, however, does not take into consideration bridge transit modes.

Research of the multi-route planning problem of multimodal transportation was done by Lue et al. [43]. In their paper, the reconstruction of lines and nodes was considered while the influencing factors and reconstruction approaches were studied. The authors proposed a route planning model and KSP algorithm, which was improved to enhance the results-finding performance. To validate the proposed algorithms, several testing networks and an empirical example based on a real scenario of oversized and heavyweight transport were calculated and analysed. The results demonstrate that the usage of reconstruction measurements can optimize the transport schemes and the proposed algorithm is capable of developing multiple transportation schemes to provide support for decision-making and risk prevention and control for the carrier. The authors are also considering the critical parameters of bridges in their algorithms, but from this paper it is not clear how to analyse and assess them.

The project approach to logistics, mainly in oversized cargo transport, was analysed by Pisz et al. in [44]. The authors identified key factors that need to be taken under consideration when planning oversized cargo transport services, such as specific characteristics, conditions, technology, etc. The proposed method uses the fuzzy set theory. The authors showed an analogy between the management of orders for oversized cargo transport services and project management. They also proposed a project approach for oversized cargo transport. Their project approach takes into consideration bridges, but no concrete approach is presented.

The authors in [45] analysed the routing stage for an oversized vehicle while having a focus from a traffic safety viewpoint. They were able to define the main factors that were determining the route for oversized vehicles: vehicle parameters, maximum single axle weight, gross vehicle weight, and atmospheric conditions. They were also emphasizing that the private carriers should have access to infrastructure data collected by the road manager. This should reduce accidents and the risk of choosing the wrong way. It is also important to build appropriate information communications technology (ICT) systems, which synergize the data of private carriers and the information collected by road administrators. Data from the road signs of road managers are not sufficient to determine bridge transit modes.

Zong et al. summarized in their research [46] the key elements for the safety of overweight/oversized cargo transport. They introduced several means to optimize the transportation process, such as the selection of trailer, truck, and platform lorry, design of lashing programs, simulation of transport, etc. They also created a framework design for overweight/oversized cargo transport, mainly in terms of transport operation.

Dolezel et al. studied the load-bearing capacity of existing bridges in their research [47]. They developed a comprehensive procedure to determine the reliability and load-bearing capacity level of the existing bridges on highways and roads. This was possible by using advanced methods of reliability analysis based on Monte Carlo-type simulation techniques in combination with a nonlinear finite element method analysis. The authors proved that

probabilistic methods in combination with nonlinear FEM analysis represent an effective and practical tool in cases for the evaluation of load-bearing capacity and reliability of existing structures. However, their model is more suitable for the static assessment of bridges and depends on large amounts of data, which are not available for global assessment.

Ghisolfi et al. [48] analysed the impact of overweight vehicles in Brazil. They were able to evaluate the effect on costs due to transportation, pavement maintenance, and road accidents using the System Dynamics method. In their study, the transport of ornamental stones was analysed. The authors proposed a model that contributes to the understanding of the dynamic behaviour of heavyweight transport in Brazil under different vehicle loading policies. The authors focused on the effects of overweight vehicles on pavements but not bridges.

Mikusova et al. determined in their research [49] the optimum size of parking places that can also be used for oversized transport in order to determine the swept envelopes for oversized vehicle combinations or for the design of parking places for such vehicle combinations.

Skrucany et al. determined in their research [50] the methodology of measurement for a vehicle's centre of gravity. The centre of gravity influences the axle loads, which are parameters that are considered for bridge assessment.

Based on the above-mentioned literature, it can be concluded that the research in this field predominantly pays attention to the following areas:

- limiting engineering constructions (intersections, bridges, parking areas): [1–11,47,49];
- optimisation of transport routes: [12–18,42,43];
- transport technology, methodology, and legislation: [25–31,44];
- transport safety and risk management: [17,32–37,41,45,46,48,50];
- oversized and excessive transport by sea: [38–40].

Relevant research questions include the following: Is it possible to realise the global assessment of passage bridges on a selected area on the basis of the data available? Is it possible to realise it after there are adjustments made in the data available?

## 2. Materials and Methods

The legislative requirements for oversized and excessive transports, load rating of bridges, and transit modes, as well as the procedure of determining a critical vehicle, shall be considered when designing the global assessment of bridges.

### 2.1. Legislative Requirements in Slovakia

In Slovakia, the exclusive use of roads relates to the Act No. 135/1961 Coll. on roads (Road Act), based on which an authorisation from the road authorities is needed—except as it may be specified in the Act—and is issued according to the opinion of the road administrator and the binding opinion of the licensing authorities within its scope pursuant to the specific rules [51].

This section describes the special usage of road infrastructure for abnormal transport and how abnormal transport is under permission for the exclusive use of roads except for the transport of vehicles used by the armed forces and armed security forces, as well as for the travel of agricultural machines and equipment when working within the farmed area. If the abnormal transport vehicle with a total mass over 60 t crosses the railway, the applicant is to require permission for railway crossing from the owner or operator of the railway infrastructure. Similarly, if an abnormal transport vehicle with a height over 4.5 m crosses under the trolley-line, the applicant is to require permission for crossing from the administrator, owner, or operator of the trolley-line. Permission for special road use can be issued for single transport or multiple transport, where single transport is performed at one day without a determined route or on a determined route on any day within 30 days from the date of notification for permission to the applicant. Multiple transport means a number of transports performed on a determined route, multiple determined, or without any determined route. The procedure for issuing the permission for the exclusive use of

roads for the transport of excessive and oversized cargo is performed according to the Technical Conditions approved by the Ministry of Transport and Construction of the Slovak Republic, such as Technical Conditions TP 103—Excessive and oversized transport [51,52].

The technical conditions cover the issue of excessive and oversized transport and include conditions and procedures needed for:

- the application for the special usage of road infrastructure for abnormal transport;
- the authorisation procedure for the special usage of road infrastructure for abnormal transport;
- performing abnormal transport as well as related activities of parties concerned;
- The inspection of vehicle masses and dimensions;
- The registration of people entitled to escort abnormal transport.

These conditions also define the selected conditions that are important for the construction and reconstruction of road infrastructure, bearing in mind excessive and oversized transport.

Point 4.11 of Technical Conditions TP 103 enables, where necessary, the use of other vehicle/vehicle combinations than the one with the permission for special use of the road in relation to oversized transport, which means that the dimensional and mass parameters of both combinations are the same, or the total mass of the vehicles/vehicle combinations and their axle load are lower after replacement.

The essential parameters in terms of loads on bridges are the masses corresponding to the axles, their number, and the wheelbase. When crossing these engineering constructions, the instantaneous maximum weight is important. That also means that the number of axles with a certain load is on the bridge. To know these parameters, it is necessary to have a model of the cumulative axle loads depending on the cumulative wheelbase.

### 2.2. Procedure of Determining a Critical Vehicle

The purpose of modelling is to determine a critical vehicle that can be later used for the global assessment of the bridges. A critical vehicle has the characteristics of all the considered vehicles/vehicle combinations for the global assessment of bridges. The number of axles, axle wheelbases, and axle load of all the considered vehicles/vehicle combinations are considered as input data. The output of the model are the cumulative wheelbases and cumulative axle loads of the critical vehicle/vehicle combination, which are then used for the global assessment of bridges.

To have similar critical values based on the input values of several combinations, the following procedure is needed:

Description of variables used in the model

- $i \in \langle 1; p \rangle$, where $p$ is the number of vehicles/vehicle combinations;
- $j \in \langle 1; n \rangle$, where $n$ is the number of vehicle/vehicle combination's axles;
- $l_{ij} \in \langle 0; l \rangle$, length of $j$-wheelbase of vehicle $i$ [m], *while l is the length of the vehicle/vehicle combination and*;
- it is applied for the first axle that $l_{i1} = 0$;
- $m_{ij}$ is the load of $j$ axle for $i$-vehicle [t];
- $m_i$ is the total mass of $i$-vehicle, $m_i \approx \sum\limits_{j=1}^{n} m_{ij}$;
- $m_i^{cum}(j)$ is the cumulative load for $j$ axles of $i$-vehicle;
- $\max m_i^{cum}(j)$ is the maximum cumulative load for $j$ axles of $i$-vehicle;
- $l_i^{cum}(j)$ is the cumulative wheelbase for $j$ axles of $i$-vehicle;
- $\min l_i^{cum}(j)$ is the minimum cumulative wheelbase for $j$ axles of $i$-vehicle;

1. Inserting the basic identification data for $n$ vehicles/vehicle combinations.
2. Setting the load $m_{ij}$ for each axle of $j$ vehicle/vehicle combination $i$.
3. Calculation of the maximum cumulative load for $j$ vehicle axles $i$: $\max m_i^{cum}(j)$.

    1. The calculation of single axle load $m_{is}^{cum}(1) = m_{is}$, $s \in \{1.2,\dots,n\}$ and the determination of maximum single axle load $\max m_i^{cum}(1)$, as a maximum value from the values calculated $m_{is}^{cum}(1)$.

2. The calculation of the load of two neighbouring axles $m_{is}^{cum}(2) = m_{i(s-1)} + m_{is}$, $s \in \{2, \ldots, n\}$ and the determination of the maximum cumulative load of two neighbouring axles $\max m_i^{cum}(2)$, as a maximum value from the values calculated $m_{is}^{cum}(2)$.

3. The calculation of the load of three neighbouring axles $m_{is}^{cum}(3) = m_{i(s-2)} + m_{i(s-1)} + m_{is}$, $s \in \{3, \ldots, n\}$ and the determination of the maximum cumulative load of three consecutive axles $\max m_i^{cum}(3)$, $m_{is}^{cum}(3)$. $m_{is}^{cum}(3)$.

$n$: The calculation of the maximum cumulative load for all the axles together, $\max m_i^{cum}(n) = m_{i1} + m_{i2} + \cdots + m_{in}$.

4. The calculation of the minimum cumulative wheelbase for $j$ axles of vehicle $i$: $\min l_i^{cum}(j)$.

   1: The determination of a single axle's wheelbase $l_i^{cum}(1) = 0 = l_{is}$.

   2: The calculation of the wheelbase of two neighbouring axles $l_{is}^{cum}(2) = l_{i(s-1)} + l_{is}$, $s \in \{2, \ldots, n\}$ and the determination of the maximum wheelbase of two neighbouring axles $\min l_i^{cum}(2)$ as a minimum value from the values calculated $l_{is}^{cum}(2)$.

   3: The calculation of the cumulative wheelbase of three consecutive axles $l_{is}^{cum}(3) = l_{i(s-2)} + l_{i(s-1)} + l_{is}$, $s \in \{3, \ldots, n\}$ and the determination of the minimum cumulative wheelbase of three consecutive axles $\min l_i^{cum}(3)$ as a minimum value from the values calculated $l_{is}^{cum}(3)$.

   $n$: The calculation of the cumulative wheelbase for all the axles together $\min l_i^{cum}(n) = l_{i1} + l_{i2} + \cdots + l_{in}$.

5. The determination of the maximum ratio for the cumulative load and relevant wheelbase $(j) = \max\{m_{is}^{cum}(j) / l_{is}^{cum}(j)\}$ for $j$ axles of vehicle/vehicle combination $i$, where $s$ designates the order number of the last axle in the order of $j$ axles.

6. The graphical display of the maximum cumulative loads for $j$ axles, $\max m_i^{cum}(j)$ depending on the minimum cumulative wheelbases of $j$ axles, and $\min l_i^{cum}(j)$ for vehicle $i$ given by a broken line.

7. The assessment of a vehicle/vehicle combination substitution The graphical display of a cumulative load depending on the wheelbase (step 5) for the given vehicle/vehicle combination $i$ and the considered substitute vehicle/vehicle combination $k$ can determine whether the broken line for substitute vehicle $k$ is south-east (under and right) of the broken line of given vehicle $i$, which means that vehicle $k$ has lower or the same axle loads as vehicle $i$, and thus, it can replace the given vehicle $i$. If the broken line for given vehicle $k$ is north-west (above or left) of the broken line of vehicle $i$, vehicle $k$ exceeds the parameters of vehicle $i$, and thus it cannot be replaced.

8. The display of the critical vehicle combination (VC) for the given set $n$ of vehicles/vehicle combinations. This critical vehicle combination is a theoretical vehicle combination. The cumulative loads, depending on the wheelbase (step 5) for each vehicle/vehicle combination $i \in \{1, \ldots, p\}$, are displayed by broken lines. The broken line in the form of a convex envelope of a set of broken lines to the north-west represents the most unsatisfactory parameters for a load, depending on the minimum cumulative wheelbase $l_i^{cum}(k)$, and the maximum cumulative axle loads $m_i^{cum}(k)$, where $k$ represents the theoretical axles of the critical vehicle/vehicle combination.

Theoretically, in this way, it is possible to adapt an infinite number of vehicles/vehicle combinations that are composed of an infinite number of axles. However, for practical needs, there are several limitations, which are given as follows:

- $m_{ij} \leq 14$ t;
- $m_i \leq 120$ t;
- $n \leq 15$.

The authors of the paper established an application through which it is possible to assess whether it is permissible to substitute a vehicle/vehicle combination that has a special use permit for driving on roads for abnormal transport.

It is necessary to have the axle loads of assessed combinations in tonnes and the wheelbases of axle pairs in metres, while for axle 1, the wheelbase is 0 m. A filled-out

sample of axle loads and wheelbases can be seen in Table 1 for 15 vehicle combinations (further as VC).

**Table 1.** Written sample of axle's wheelbases.

| Vehicle Combination | Axle Load—$m_{ij}$ | | | | | | | | | | | | | | Sum of Axle Loads |
|---|---|---|---|---|---|---|---|---|---|---|---|---|---|---|---|
| | 1 | 2 | 3 | 4 | 5 | 6 | 7 | 8 | 9 | 10 | 11 | 12 | 13 | 14 | |
| VC 1 | 9.30 | 10.85 | 10.85 | 10.40 | 10.40 | 10.40 | 10.40 | 10.40 | | | | | | | 83.00 |
| VC 2 | 7.80 | 11.40 | 11.40 | 11.40 | 11.40 | 11.40 | 11.40 | 11.40 | 11.40 | | | | | | 99.00 |
| VC 3 | 7.55 | 7.55 | 7.55 | 7.55 | 10.36 | 10.36 | 10.36 | 10.36 | 10.36 | 10.36 | | | | | 92.36 |
| VC 4 | 5.00 | 7.00 | 8.00 | 8.00 | 9.00 | 9.00 | 9.00 | 9.00 | 9.00 | 9.00 | 9.00 | 9.00 | | | 100.00 |
| VC 5 | 12.00 | 12.00 | 12.00 | 12.00 | 12.00 | 12.00 | 12.00 | 12.00 | | | | | | | 96.00 |
| VC 6 | 5.00 | 7.00 | 8.00 | 8.00 | 9.00 | 9.00 | 9.00 | 9.00 | 9.00 | 9.00 | 9.00 | 9.00 | | | 100.00 |
| VC 7 | 7.50 | 7.50 | 10.00 | 10.00 | 10.00 | 10.00 | 10.00 | 10.00 | 10.00 | 10.00 | 10.00 | 10.00 | | | 115.00 |
| VC 8 | 8.00 | 7.50 | 7.50 | 7.50 | 7.50 | 7.50 | 7.50 | 8.00 | 7.00 | 7.74 | | | | | 74.74 |
| VC 9 | 8.00 | 8.00 | 8.00 | 8.00 | 8.00 | 8.00 | 8.00 | 8.00 | 8.00 | | | | | | 72.00 |
| VC 10 | 6.00 | 8.00 | 9.00 | 10.00 | 9.00 | 9.00 | 9.00 | | | | | | | | 60.00 |
| VC 11 | 7.00 | 8.00 | 9.50 | 9.50 | 9.60 | 9.60 | 9.60 | 9.60 | 9.60 | | | | | | 82.00 |
| VC 12 | 5.00 | 8.00 | 10.00 | 9.50 | 9.50 | 9.50 | 9.50 | 9.50 | 9.50 | 9.50 | 9.50 | | | | 99.00 |
| VC 13 | 5.00 | 7.00 | 8.00 | 8.00 | 7.00 | 7.00 | 7.00 | 7.00 | 7.00 | 7.00 | 7.00 | 7.00 | 7.00 | 7.00 | 98.00 |
| VC 14 | 6.65 | 9.50 | 9.50 | 9.50 | 9.50 | 9.50 | 9.50 | 9.50 | | | | | | | 73.15 |
| VC 15 | 7.50 | 7.50 | 10.50 | 10.50 | 10.50 | 10.50 | 10.50 | 10.50 | 10.50 | 10.50 | 10.50 | 10.50 | | | 120.00 |

| Vehicle Combination | Wheelbase $l_{ij}$ | | | | | | | | | | | | | | Sum of Wheelbases |
|---|---|---|---|---|---|---|---|---|---|---|---|---|---|---|---|
| | 1 | 2 | 3 | 4 | 5 | 6 | 7 | 8 | 9 | 10 | 11 | 12 | 13 | 14 | |
| VC 1 | 0.00 | 3.45 | 1.35 | 5.60 | 1.31 | 1.31 | 1.31 | 1.31 | | | | | | | 15.64 |
| VC 2 | 0.00 | 2.60 | 1.36 | 1.36 | 1.36 | 1.36 | 1.36 | 1.36 | 1.36 | | | | | | 21.26 |
| VC 3 | 0.00 | 1.65 | 2.87 | 1.45 | 2.30 | 1.40 | 1.40 | 1.40 | 1.40 | 1.40 | | | | | 15.27 |
| VC 4 | 0.00 | 2.55 | 1.35 | 1.35 | 3.70 | 1.36 | 1.36 | 7.47 | 1.36 | 1.36 | 1.36 | 1.40 | | | 24.62 |
| VC 5 | 0.00 | 1.60 | 2.80 | 1.60 | 1.60 | 1.60 | 1.60 | 1.60 | | | | | | | 12.40 |
| VC 6 | 0.00 | 2.55 | 1.35 | 3.70 | 1.36 | 1.36 | 7.47 | 1.36 | 1.36 | 1.36 | 1.36 | 1.36 | | | 24.59 |
| VC 7 | 0.00 | 2.60 | 1.45 | 1.41 | 5.00 | 1.41 | 1.41 | 1.41 | 1.41 | 1.41 | 1.41 | 1.41 | | | 20.33 |
| VC 8 | 0.00 | 2.63 | 1.46 | 1.46 | 2.24 | 1.36 | 7.80 | 1.36 | 1.36 | 1.36 | | | | | 21.03 |
| VC 9 | 0.00 | 2.59 | 1.36 | 1.36 | 28.00 | 1.36 | 1.36 | 1.36 | 1.36 | | | | | | 38.75 |
| VC 10 | 0.00 | 2.40 | 1.33 | 11.20 | 1.36 | 1.36 | 1.36 | | | | | | | | 19.01 |
| VC 11 | 0.00 | 2.28 | 1.32 | 1.36 | 7.14 | 1.81 | 1.81 | 1.81 | 1.81 | | | | | | 19.34 |
| VC 12 | 0.00 | 3.40 | 1.37 | 3.50 | 1.55 | 1.55 | 1.55 | 1.55 | 1.55 | 1.55 | 1.55 | | | | 19.12 |
| VC 13 | 0.00 | 2.55 | 1.35 | 1.35 | 3.10 | 1.50 | 1.50 | 1.50 | 1.50 | 1.50 | 1.50 | 1.50 | 1.50 | 1.50 | 21.85 |
| VC 14 | 0.00 | 3.60 | 1.40 | 4.50 | 1.36 | 1.36 | 1.36 | 1.36 | | | | | | | 14.94 |
| VC 15 | 0.00 | 2.60 | 1.44 | 1.42 | 2.60 | 1.50 | 1.50 | 12.40 | 1.50 | 1.50 | 1.50 | 1.50 | | | 29.46 |

The application further displays the graphs of the cumulative loads depending on the wheelbases for particular vehicles/vehicle combinations, as seen in Figure 1:

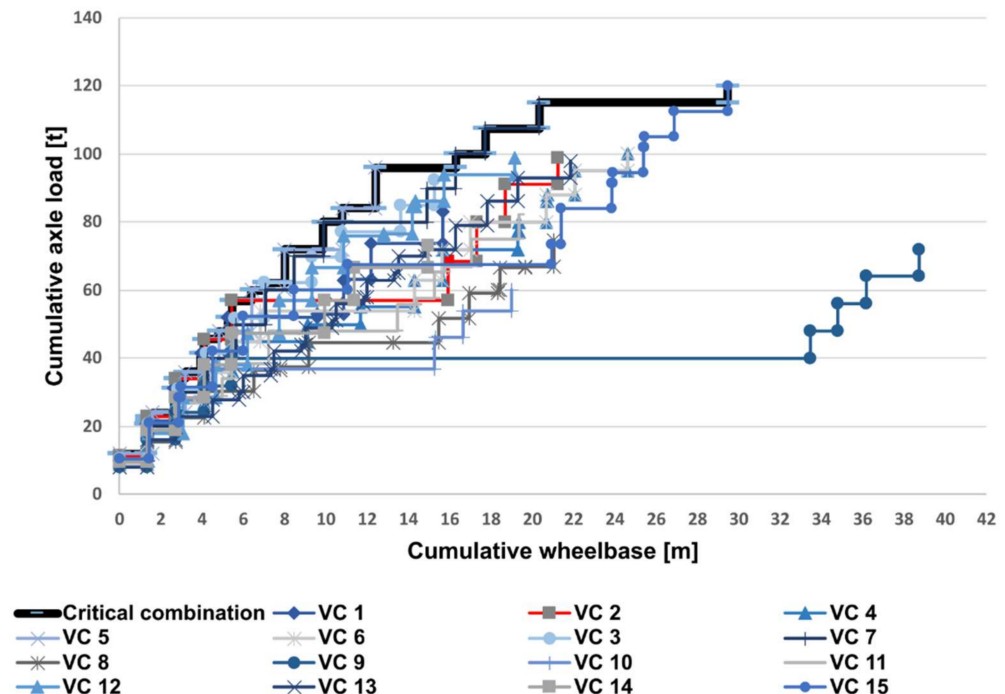

**Figure 1.** Sample of graphs of cumulative axle loads depending on the wheelbase for 15 vehicle combinations (VC).

## 3. Results

The examples of assessing whether it is possible to replace the vehicles/vehicle combinations by other vehicles/vehicle combinations are shown in the model examples, and the input parameters of the vehicle combinations are given in Table 2. The vehicle combinations are split into three groups. The vehicle combinations from VC 16 to VC 18 have the same axle loads and different wheelbases. The vehicle combinations from VC 19 to VC 21 have the same axle wheelbases but different axle loads. The vehicle combinations from VC 22 to VC 24 have different wheelbases and different axle loads. Every vehicle combination group is assessed individually and further described as model examples 1, 2, and 3. The parameters $l_{ij}$ and $m_{ij}$ of the semi-trailer vehicle combinations are in Figure 2 and the parameter values are in Table 2.

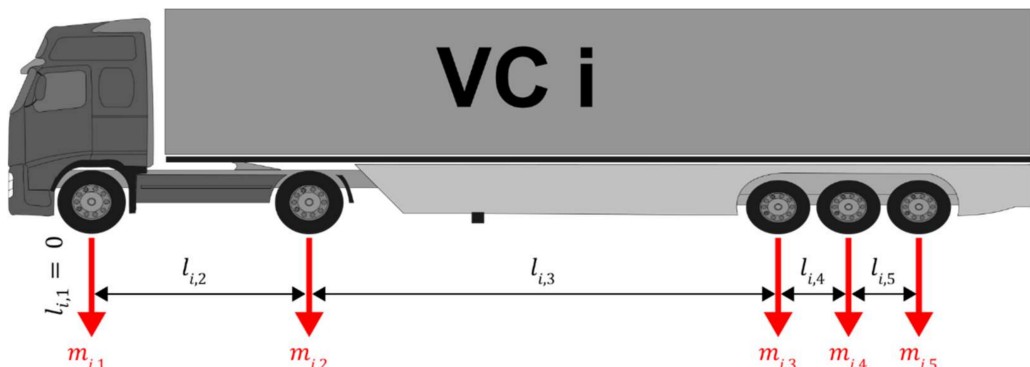

**Figure 2.** Parameters $l_{ij}$ and $m_{ij}$ of semi-trailer vehicle combinations used in model examples 1–3.

### 3.1. Model Example 1

The first example can be represented by a three five-axle vehicle combination with the same loads on the axles, and these combinations differ in their wheelbases between 2 and 3 axle, 3 and 4 axle, and between their 4 and 5 axle. The route is assessed for

combination 16. There is a question whether it is possible to substitute combination 16 with combination 17 or combination 18, in accordance with point 4.11 TP103 of the Technical Conditions. The parameters of the combinations are given in Table 2. Taking into account the cumulative axle load depending on the wheelbase in the graph in Figure 3, it can be seen that all the graph points of combination 17 show a lower or the same axle load than combination 16, which means that it is suitable from the replacing conditions' point of view. On the contrary, combination 18 exceeds the parameters of combination 16, which means that it cannot be substituted with combination 16 in accordance with point 4.11 TP 103 in the Technical Conditions.

**Table 2.** Parameters of vehicle combinations (VC) assessed.

| | | Order of Axles—*j* | | | | | Sum |
|---|---|---|---|---|---|---|---|
| | | 1 | 2 | 3 | 4 | 5 | |
| VC 16 | wheelbase$l_{16,j}$ [m] | 0.00 | 3.70 | 5.90 | 1.31 | 1.41 | 12.32 |
| | axle load$m_{16,j}$ [t] | 8.00 | 11.50 | 8.50 | 8.50 | 8.50 | 45.00 |
| VC 17 | wheelbase$l_{17,j}$ [m] | 0.00 | 3.70 | 6.00 | 1.41 | 1.41 | 12.52 |
| | axle load$m_{17,j}$ [t] | 8.00 | 11.50 | 8.50 | 8.50 | 8.50 | 45.00 |
| VC 18 | wheelbase$l_{18,j}$ [m] | 0.00 | 3.70 | 5.50 | 1.31 | 1.31 | 11.82 |
| | axle load$m_{18,j}$ [t] | 8.00 | 11.50 | 8.50 | 8.50 | 8.50 | 45.00 |
| VC 19 | wheelbase$l_{19,j}$ [m] | 0.00 | 3.70 | 5.90 | 1.31 | 1.41 | 12.32 |
| | axle load$m_{19,j}$ [t] | 8.00 | 11.50 | 8.50 | 8.50 | 8.50 | 45.00 |
| VC 20 | wheelbase$l_{20,j}$ [m] | 0.00 | 3.70 | 5.90 | 1.31 | 1.41 | 12.32 |
| | axle load$m_{20,j}$ [t] | 8.00 | 11.00 | 7.50 | 7.50 | 7.50 | 41.50 |
| VC 21 | wheelbase$l_{21,j}$ [m] | 0.00 | 3.70 | 5.90 | 1.31 | 1.41 | 12.32 |
| | axle load$m_{21,j}$ [t] | 8.00 | 12.00 | 9.50 | 9.50 | 9.50 | 48.50 |
| VC 22 | wheelbase$l_{22,j}$ [m] | 0.00 | 3.70 | 5.90 | 1.31 | 1.41 | 12.32 |
| | axle load$m_{22,j}$ [t] | 8.00 | 11.50 | 8.50 | 8.50 | 8.50 | 45.00 |
| VC 23 | wheelbase$l_{23,j}$ [m] | 0.00 | 3.70 | 5.50 | 1.31 | 1.31 | 11.82 |
| | axle load$m_{23,j}$ [t] | 8.00 | 11.00 | 7.50 | 7.50 | 7.50 | 41.50 |
| VC 24 | wheelbase$l_{24,j}$ [m] | 0.00 | 3.70 | 6.00 | 1.41 | 1.41 | 12.52 |
| | axle load$m_{24,j}$ [t] | 8.00 | 12.00 | 9.50 | 9.50 | 9.50 | 48.50 |

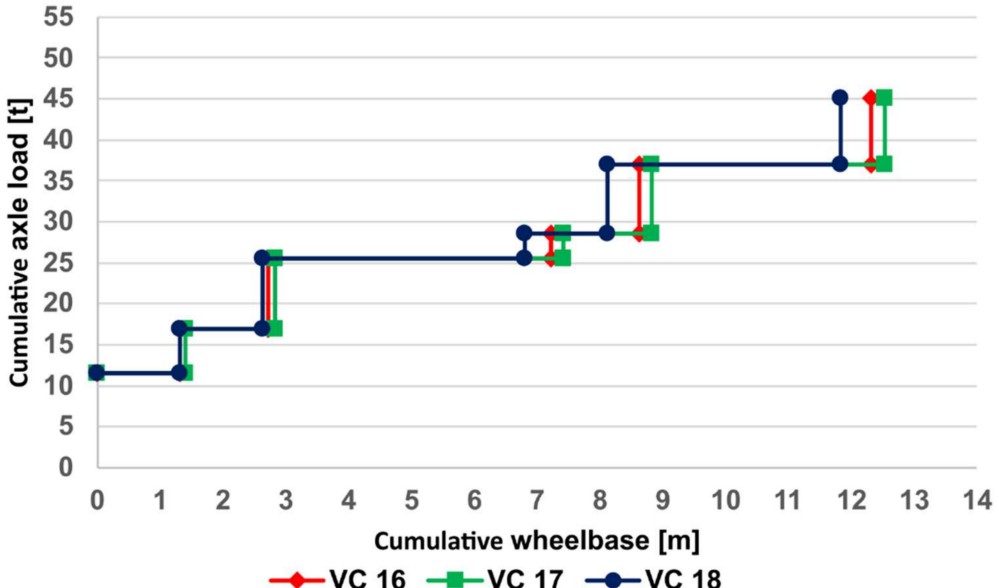

**Figure 3.** Graphical comparison of cumulative load depending on the wheelbase at same axle loads.

### 3.2. Model Example 2

The second example can be represented by a three five-axle vehicle combination with the same axle wheelbases, and these combinations differ in the 2, 3, 4, and 5 axle loads. The parameters of these combinations are given in Figure 4. Again, it is necessary to find out whether it is possible to substitute combination 19 with combination 20 or combination 21, in accordance with point 4.11 TP103 of the Technical Conditions.

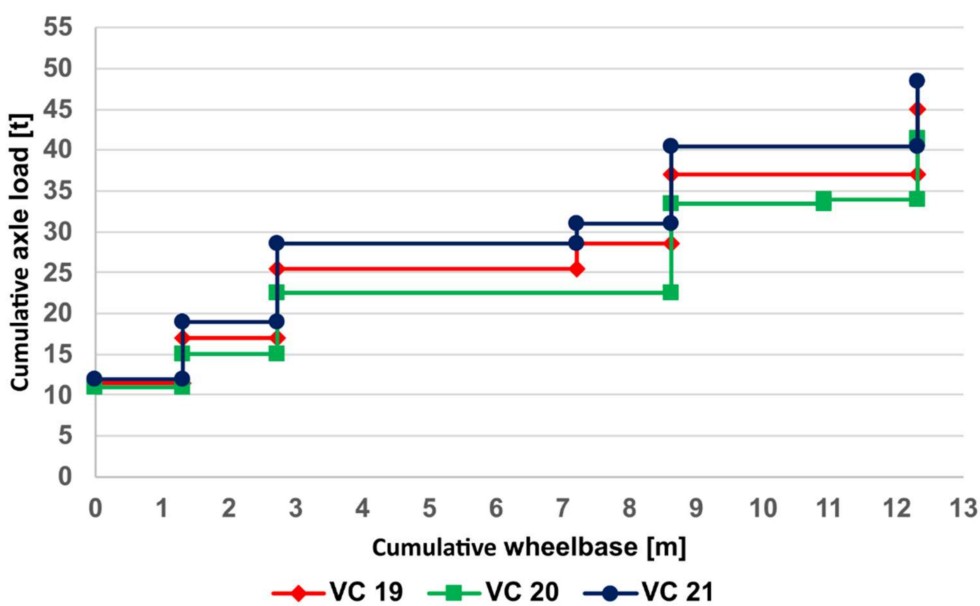

**Figure 4.** Graphical comparison of cumulative load depending on the wheelbase at same axle wheelbase.

Taking the cumulative axle load into account, and depending on the wheelbase in the graph in Figure 3, it can be seen that all the graph points of combination 20 show a lower axle load than combination 19, which means that it is suitable from the replacing conditions' point of view, according to point 4.11. Combination 21 has higher loads in all the assessed views, which means that it cannot be substituted with combination 19 in accordance with point 4.11 TP 103 of the Technical Conditions.

### 3.3. Model Example 3

The third example can be represented by the comparison of two vehicle combinations that differ in the wheelbase between the 2 and 3 axle and between the 3 and 4 axle and in axle loads 2, 3, 4, and 5, of which the parameters are given in Table 2.

It can be seen in the graph in Figure 5 that combination 22 cannot be substituted with either combination 23 or combination 24, in accordance with point 4.11 TP 103 of the Technical Conditions, since combination 23 exceeds the axle load of combination 22 with a bridge length <2.62; 2.72) m, <8.12; 8.62) m, and <11.82; 12.32) m. Combination 24 exceeds the axle load of combination 22 with a bridge length lower than 1.31 m as well as with lengths <1.41; 2.72) m, <2.82; 7.21) m, <7.41; 8.62) m, <8.82; 12.32) m, and with length 12.52 m or more.

The application, in the last step, determines the parameters of a critical vehicle combination, i.e., its cumulative wheelbases and axle loads as well as the axle loads used as input data for the particular route assessment, or for the global assessment of bridge passage. For assessing the route in relation to transport with a total mass up to 60 t and a height that does not exceed 4.5 m, it can be used as an application for the assessment of bridge passage, which is available on the website of the Slovak Road Administration (further as SRA) and is used by inserting the input data in the form of partial wheelbases and all the axle loads of the critical vehicle combination. However, relating to the global assessment, there was use for the data on cumulative wheelbases and axle loads that was suggested, since this

considers their own assessment of the bridge passage mode based on the "raw data" on bridge $b$, which includes:

- $d(b)$ length of bridging [m];
- $V_n(b)$ normal load [t];
- $V_r(b)$ exclusive load [t];
- $V_e(b)$ exceptional load [t].

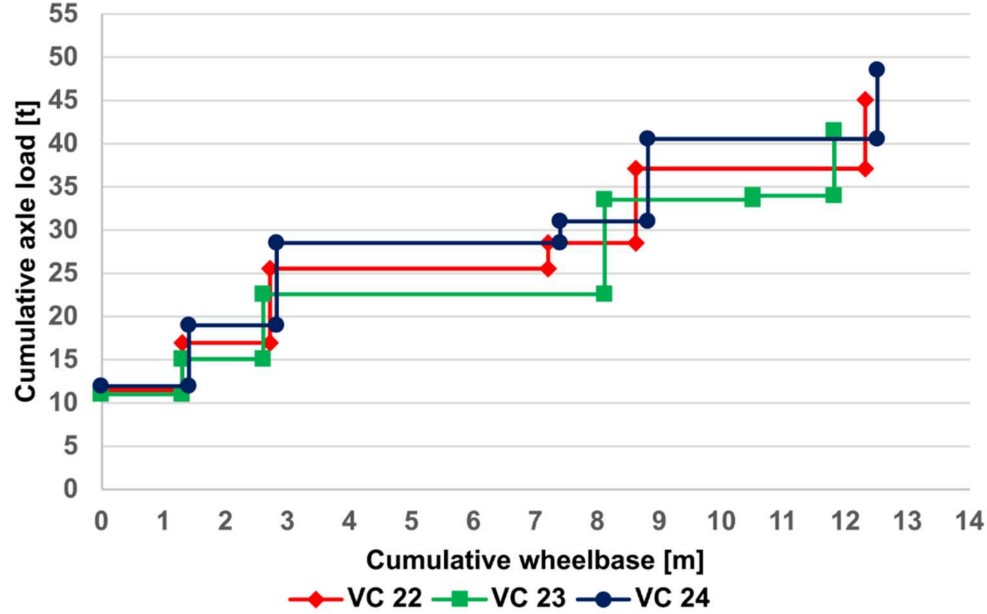

**Figure 5.** Graphical comparison of cumulative load depending on the wheelbase at different axle wheelbases and loads.

The global assessment of bridge passage will include the evaluation of the maximum bridge load caused by a vehicle/vehicle combination via the comparison of the maximum axle loads on the bridge to the normal, exclusive, and exceptional loads. If the maximum cumulative axle load $m_i^{cum}(k)$ on the bridge is larger than the exceptional load of the bridge $V_e(b)$, a static assessment of the bridge will be required. If the exclusive load $V_r(b)$ is exceeded, and the exceptional $V_e(b)$ one is not exceeded, it is necessary for the vehicle/vehicle combination to cross the bridge under the conditions given for the exceptional transit mode, i.e., in the ideal trace (in the middle of bridge) at the speed of maximum 5 km/h without pushing. If the value of the normal load $V_n(b)$ is exceeded, and the exclusive $V_r(b)$ is not, the vehicle/vehicle combination can cross the bridge under the conditions given for the exclusive transit mode, i.e., in the arbitrary trace without speed limitation, but it must be single on the bridge. If the value of the normal load $V_n(b)$ is not exceeded, the bridge passage is not limited.

The assessment of the vehicles/vehicle combinations crossing bridge $b$ with the length $d$:

- if $l_i^{cum}(k) \leq d(b)$ applies to cumulative wheelbase $k$ of the axles, then the maximum value of cumulative axle load $m_i^{cum}(k)$ at the corresponding wheelbase is compared to permitted bridge load ratings $V_n(b)$, $V_r(b)$, $V_e(b)$ :;
- if $m_i^{cum}(k) \leq V_n(b)$, transport without restriction;
- if $V_n(b) \leq m_i^{cum}(k) \leq V_r(b)$, it is necessary to have an exclusive transit mode;
- if $V_r(b) \leq m_i^{cum}(k) \leq V_e(b)$, it is necessary to have an exceptional transit mode;
- if $m_i^{cum}(k) \geq V_e(b)$, it is necessary to have a static assessment of the bridge.

The process diagram for bridge assessment is given in Figure 6.

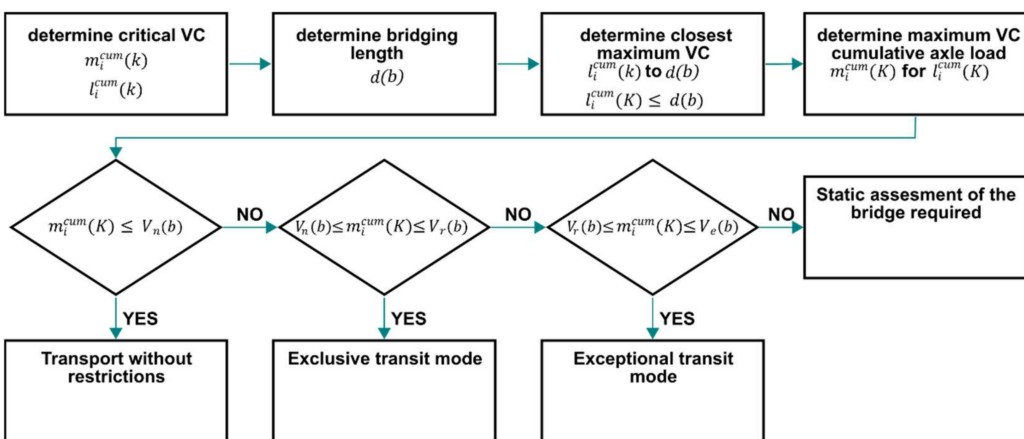

**Figure 6.** Process diagram of individual bridge assessment.

*3.4. Global Assessment*

Currently—not only in Slovakia but also in other states—the assessment of bridge passage for certain routes is used for heavy and oversized transport. This means that if we use 100 transports, 100 assessments of the individual routes are needed, even if they are the same routes or the same vehicles/vehicle combinations used for a number of transports. Roads are full of different types of vehicles/vehicle combinations for which the axle loads and distances of the axles (wheelbases) are important. There were vehicle/vehicle combination parameters analysed for which the routes in relation to heavy and oversized transport in Slovakia were assessed from 1 January 2016 to 31 December 2020. However, the file contained only data on transports with a total mass that exceeded 60 t or with a total width and/or height that exceeded 4.5 m, which means that a high number of transports with a total mass of 60 t and a width and height of 4.5 m were filtered out, and SRA has not registered them by 2021. Nevertheless, during the revision of Technical Conditions TP 103, there was a request from revisers that was accepted by the road traffic authorities to send data from applications for permission for the exclusive use of a road on a monthly basis to the Ministry of Transport and Construction of the Slovak Republic so that the need for more comprehensive analyses and processes of statistical outputs could be met. As well, this would be fundamental for the relevant design of roads and engineering, as well as any constructions or reconstructions of roads. Our analysed data consisted of information on 1859 route assessments for oversized and excessive transport, within which 932 unique vehicles/vehicle combinations were assessed. In Figure 7, we can see the variability in the parameters of vehicles/vehicle combinations used and in Figure 8 in different intervals of mass.

Based on the large variance in total mass, vehicles and vehicle combinations were divided into six categories and the numbers of transports assessed, as well as the number of unique vehicles/vehicle combinations within particular categories, were given in Table 3.

**Table 3.** Division of transports assessed into categories according to total mass.

| Interval of Mass [t] | Number of Unique Vehicles/ Combination Vehicles | Number of Transports Assessed by SRA, Road Databank | Percentage of Unique Combinations per Overall Number of SRA Assessed Transports |
|---|---|---|---|
| (0; 40> | 129 | 204 | 63.20% |
| (40; 60> | 129 | 229 | 56.30% |
| (60; 120> | 495 | 1160 | 42.70% |
| (120; 180> | 111 | 165 | 67.30% |
| (180; 240> | 45 | 74 | 60.80% |
| >240 | 23 | 27 | 85.20% |
| Total | 932 | 1859 | 50.13% |

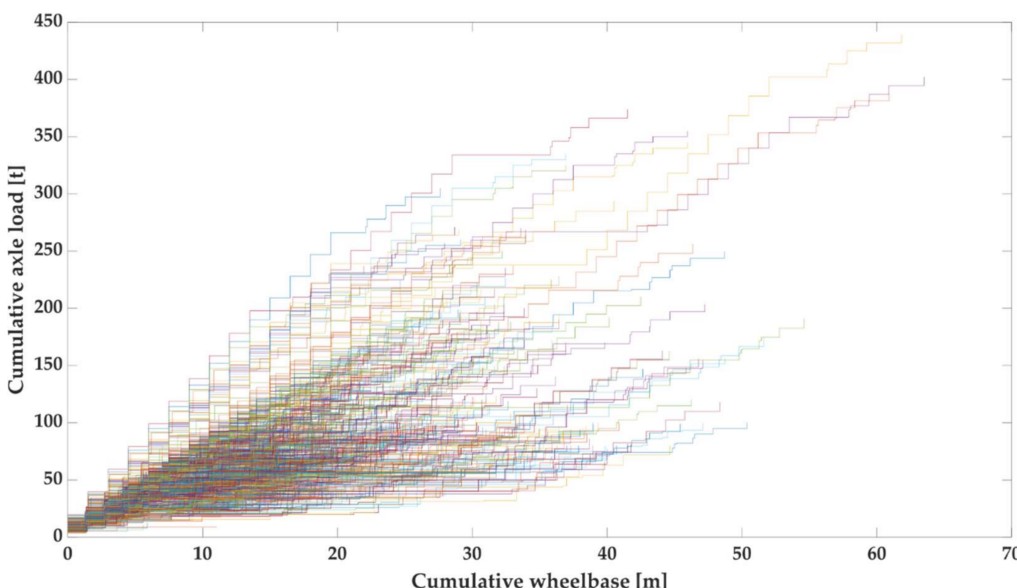

**Figure 7.** Dependence of cumulative axle load on cumulative wheelbase for 932 unique vehicles/combination vehicles showed by different colours (1859 transports).

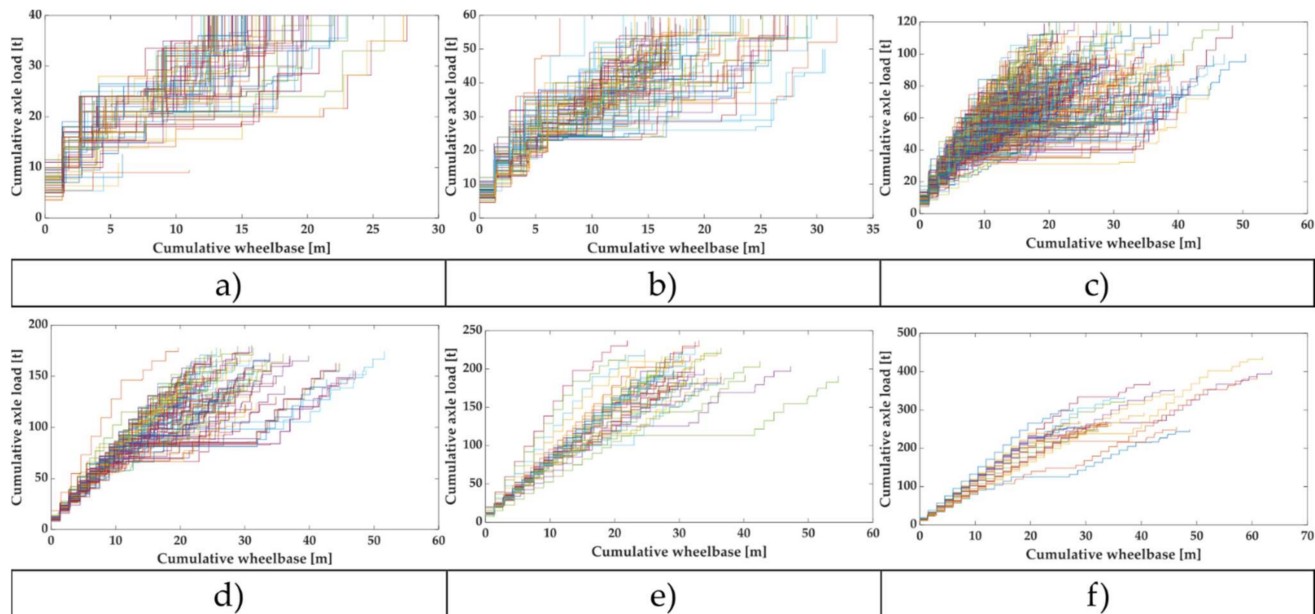

**Figure 8.** Dependence of cumulative axle load on cumulative wheelbase for 932 unique vehicles/combination vehicles showed by different colours (1859 transports) divided into intervals of mass (**a**—(0; 40 t>, **b**—(40 t; 60 t>, **c**—(60 t; 120 t>, **d**—(120 t; 180 t>, **e**—(180 t; 240 t>, **f**—>240 t).

For each interval of mass, a critical vehicle/vehicle combination was created for which all the values of marginal cumulative wheelbases and cumulative masses of axles were given. To apply these combinations into a currently used application for determining the bridge passage method, or to insert the parameters of the critical vehicle/vehicle combination into the application for a special road use permit, it is also necessary to have the partial wheelbases and partial masses of the axles, given that this was used by the global assessment of bridge passage.

Transports up to 120 t represented 85% of all the assessed transports in Table 3. A significant part of these transports is the regular transportation of construction equipment. As a model example for the global assessment of bridge passage procedure, a vehicle

combination, VC 25, was chosen with partial wheelbases and axle loads, as given in Figure 9. VC 25 is a real vehicle combination for the transport of construction equipment. VC 25 is an example of a 12-axle vehicle combination with a gross mass up to 120 t.

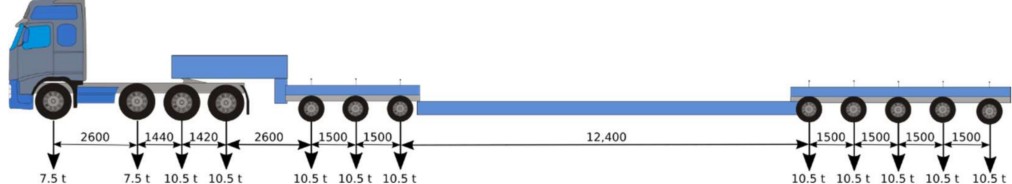

**Figure 9.** Axle loads and partial wheelbases for selected combination VC 25.

The cumulative loads were determined, i.e., the maximum axle load, respective of the group of axles with the lowest possible wheelbase and in the order in which the consecutive axles were found. It can be also seen from Figure 10 that the maximum load of a single axle with the wheelbase is equal to 0 metres and reaches a value of 10.5 t because this value is the maximum load for all single axles. If we consider a critical situation with a couple of axles, this would be composed of axles number 3 and 4 with a 21-t load and a 1.42 m partial wheelbase, and the couple would also have the lowest partial wheelbase within the combination, and, at the same time, the maximum possible load assigned to the couple of axles. A 21-t load is also reached by each couple of consecutive axles 4 + 5; 5 + 6; 6 + 7; 7 + 8; 8 + 9; 9 + 10; 10 + 11; and 11 + 12, however, all these couples have a higher partial wheelbase and are convenient from the perspective of bridge passage and road driving in general. To find out the next critical value, the smallest distance of three consecutive axles and their maximum possible load with the smallest wheelbase need to be considered. The first condition is met by the axles 2 + 3 + 4, which have their cumulative wheelbase of 2.86 m and a load of 28.5 t. The second condition is met by the axles 5 + 6 + 7, or any of the axles 8 + 9 + 10; 9 + 10 + 11, and 10 + 11 + 12, with a cumulative wheelbase of 3.00 m. The graph of dependence for the cumulative load on cumulative wheelbase is given in Figure 10.

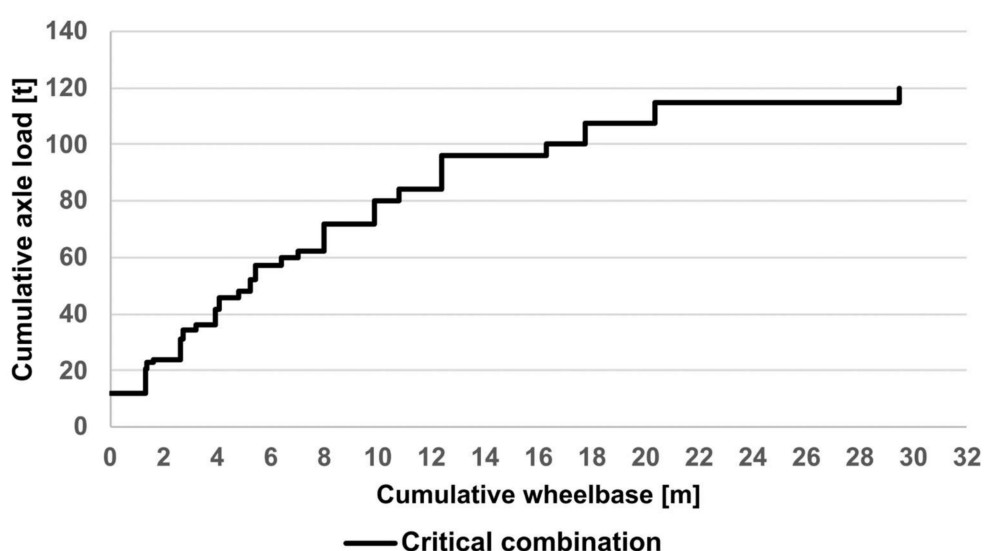

**Figure 10.** Dependence of cumulative load on cumulative wheelbase of combination VC 25.

The model results are from the cumulative axle wheelbases and their corresponding cumulative axle loads, as well as the partial wheelbases with their partial axle loads that are given in Table 4. VC 25 has 12 axles, but the theoretical critical vehicle combination created from VC 25 has 24 theoretical axles, which represent 24 different cumulative axle loads

$m_i^{cum}(k)$ and cumulative wheelbases $l_i^{cum}(k)$. The cumulative wheelbases represent interval boundaries that are later compared with the length of bridge $d(b)$ (see also Figure 6).

**Table 4.** Parameters of critical combination based on data on VC 25.

| Theoretical Axle | Cumulative Wheelbase [m] | Cumulative Axle Load [t] | Partial Wheelbase [m] | Partial Axle Load [t] | Theoretical Axle | Cumulative Wheelbase [m] | Cumulative Axle Load [t] | Partial Wheelbase [m] | Partial Axle Load [t] |
|---|---|---|---|---|---|---|---|---|---|
| 1 | 0 | 10.5 | 0 | 10.5 | 13 | 8.46 | 60 | 0.71 | 3 |
| 2 | 1.35 | 16 | 1.35 | 5.5 | 14 | 9.3 | 66.5 | 0.84 | 6.5 |
| 3 | 1.36 | 19 | 0.01 | 3 | 15 | 10.85 | 76 | 1.55 | 9.5 |
| 4 | 1.42 | 21 | 0.06 | 2 | 16 | 12.8 | 76.5 | 1.95 | 0.5 |
| 5 | 2.7 | 23 | 1.28 | 2 | 17 | 14.17 | 84.5 | 1.37 | 8 |
| 6 | 2.72 | 28.5 | 0.02 | 5.5 | 18 | 14.35 | 86 | 0.18 | 1.5 |
| 7 | 3 | 31.5 | 0.28 | 3 | 19 | 15.72 | 94 | 1.37 | 8 |
| 8 | 4.08 | 38 | 1.08 | 6.5 | 20 | 19.12 | 99 | 3.4 | 5 |
| 9 | 4.5 | 42 | 0.42 | 4 | 21 | 25.36 | 102 | 6.24 | 3 |
| 10 | 5.44 | 47.5 | 0.94 | 5.5 | 22 | 25.42 | 105 | 0.06 | 3 |
| 11 | 6 | 52.5 | 0.56 | 5 | 23 | 26.86 | 112.5 | 1.44 | 7.5 |
| 12 | 7.75 | 57 | 1.75 | 4.5 | 24 | 29.46 | 120 | 2.6 | 7.5 |

For the above-mentioned combination, the global assessment of passage of 8133 bridges registered in the database was used with the following results:

0—standard transit mode (without restriction);

1—exclusive transit mode (single vehicle on the bridge, using any trace without speed restriction);

2—exceptional transit mode (single vehicle on the bridge, using an ideal trace at the speed maximum of 5 km/h);

3—static assessment of bridge required;

4—missing data on bridge load capacity or bridging length.

In terms of the construction-technical conditions of the bridges, they can be divided into seven categories which are, together with the proportion of bridges, given in Table 5.

**Table 5.** Categorisation of 8133 bridges registered in Slovakia (authors based on [53,54]) based on construction-technical condition code (further as CTC).

| CTC | Construction-Technical Condition—Name | Description | Number of Objects | Proportion of the Overall Number |
|---|---|---|---|---|
| 1 | flawless | without any hidden or evident malfunctions | 499 | 6.14% |
| 2 | very good | design malfunctions only without affecting the load | 770 | 9.47% |
| 3 | good | larger malfunctions without affecting the load | 2314 | 28.45% |
| 4 | satisfactory | malfunctions without immediate effect on the load, but they can affect it in the future | 3033 | 37.29% |
| 5 | bad | malfunctions that have an adverse effect on the load, but they can be eliminated without replacement of faulty parts | 983 | 12.09% |
| 6 | very bad | malfunctions that affect the load and cannot be eliminated without replacement of faulty parts or replenishment of missing parts | 456 | 5.61% |
| 7 | emergency | malfunctions that affect the load and need immediate repair in order to avoid a disaster | 23 | 0.28% |
| | missing data | - | 55 | 0.68% |
| | Total | - | 8133 | 100% |

Data on the global assessment of bridge passage are displayed on the OpenStreetMap data map after its creation. Considering that, according to Technical Conditions TP 103,

if the exclusive load of any bridge is exceeded and the bridge has the CTC 6, the road infrastructure administrator can require the applicant to have two special escort vehicles registered in Slovakia, and, therefore, the figures for the global assessment of bridges are divided into two parts: the first one is displayed in Figure 11 and shows the bridges with CTC from 1 to 5 and the second part, displayed in Figure 12, shows the bridges with CTC from 6 and 7—or with missing CTC. Bridges with CTC 7 are generally not used for excessive transports.

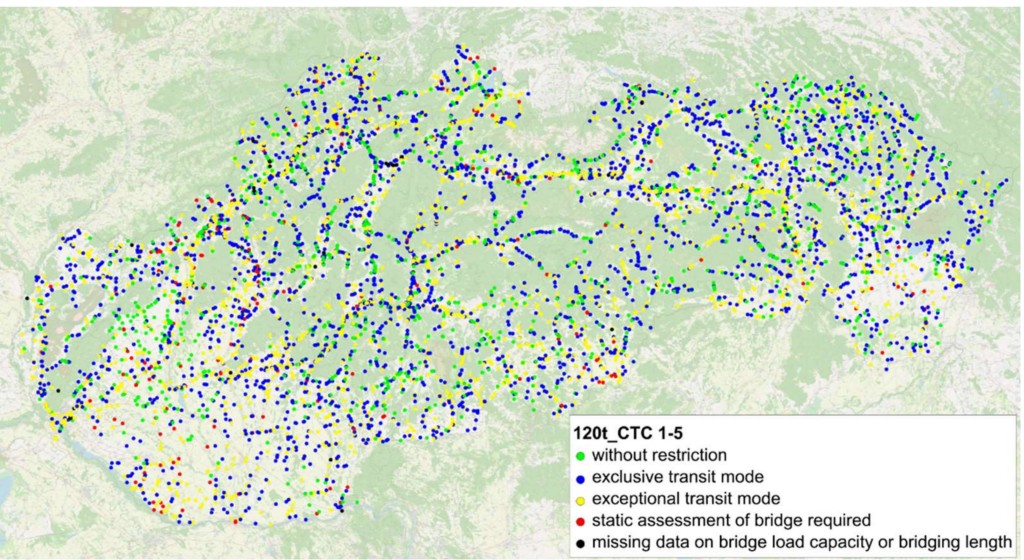

**Figure 11.** Graphical display of bridge passage with CTC 1–5 by vehicle combination VC 25 on OpenStreetMap layer (7321 bridges).

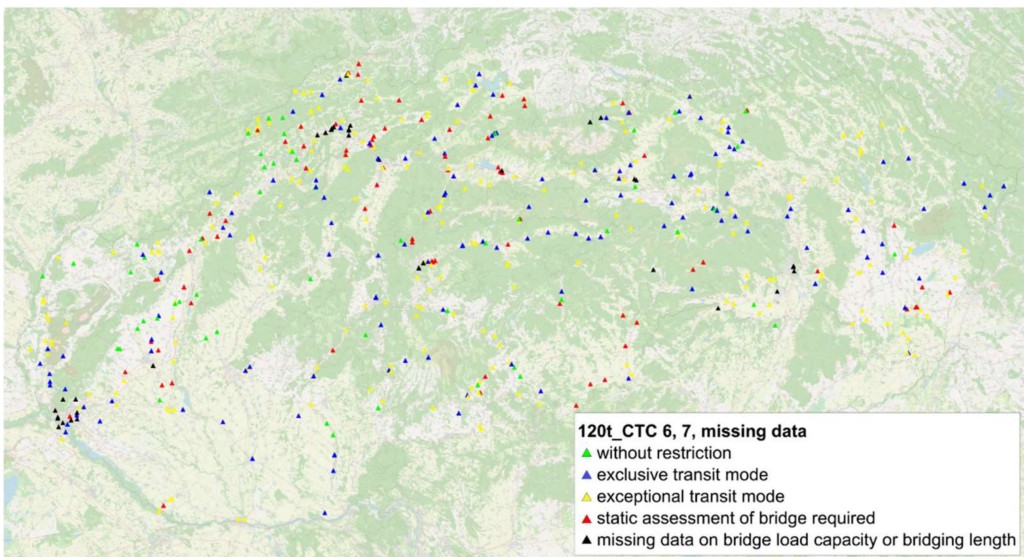

**Figure 12.** Graphical display of bridge passage with CTC 6–7 by vehicle combination VC 25 on OpenStreetMap layer (812 bridges).

Subsequently, the data were used as an input for a mobile application. The application before, or during the use of oversized loads, enables us to visually determine the way for bridge passage that needs to be observed. Based on the parameters given in advance, the application offers the option for voice announcement to inform any bridges nearby as well.

The display of bridges on the application map is based on OpenStreetMap data. Examples of bridges around the city of Žilina is given in Figure 13.

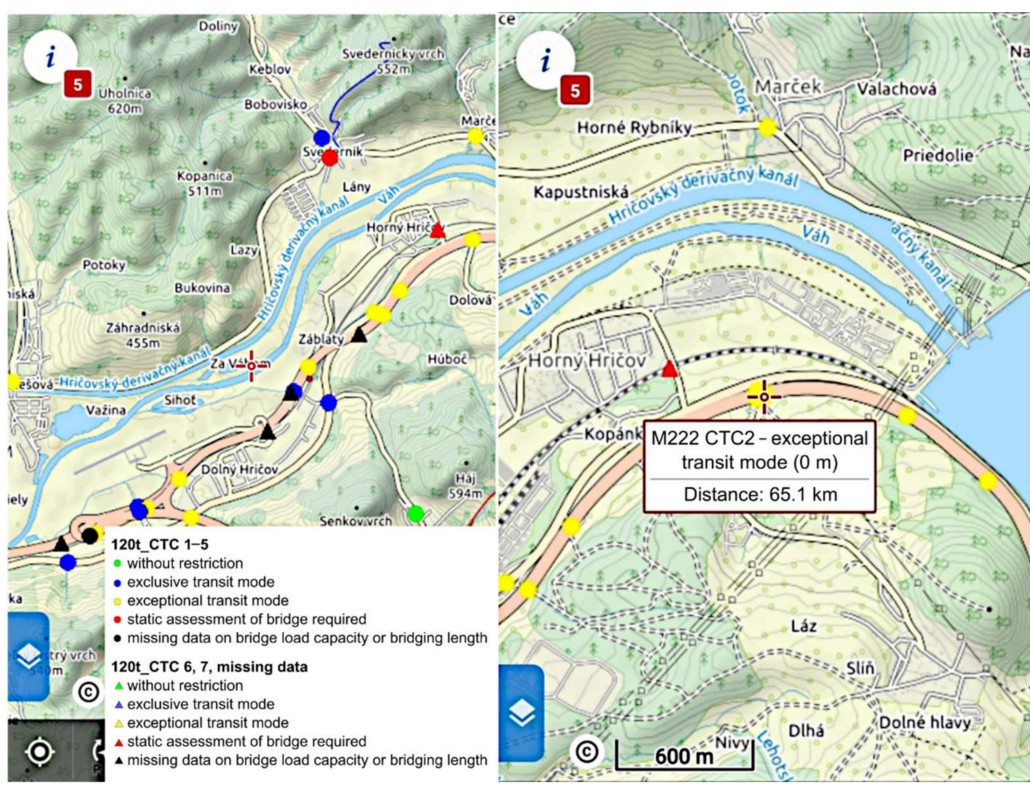

**Figure 13.** Display of bridge passage for combination VC 25 in navigational application on Open-StreetMap layer.

The display above can serve as a navigation tool with a voice announcement option to cross bridges intended either for a driver of an oversized cargo vehicle/vehicle combination, or for drivers of escort vehicles who secure passage conditions through observation.

Global assessment in Slovakia will consist of the following steps:

1.  the haulier selects a vehicle/vehicle combination, or a set of vehicles/vehicle combinations for which the global assessment shall be made;
2.  the haulier writes the parameters of these vehicles/vehicle combinations into the application form for the global assessment of the transport routes;
3.  the haulier writes the period of time for which the global assessment of the transport routes is required, 12 months maximally;
4.  SRA, Road Databank determines, according to the parameters, the critical cumulative loads and cumulative wheelbases for each of the vehicles/vehicle combinations, or for all of them;
5.  SRA, Road Databank determines the way of passage for the selected vehicle/vehicle combination, or the critical vehicle/vehicle combination for the road network of Slovakia;
6.  SRA, Road Databank provides conditions for bridge passage to the haulier in .csv and/or .xlsx format, and/or as a single map layer in .shp format;
7.  the haulier is obliged to observe the given conditions of the bridge passage every time and is obliged to use a navigation tool displaying all the transport conditions required;
8.  the results of the global assessment are applicable in the period determined by the haulier over the application for 12 months maximally, for single or multiple transport used by the selected vehicle/vehicle combination or set by the vehicles/vehicle combinations;
9.  the results of the global assessment, including the parameters of vehicles/vehicle combinations and registration numbers of all vehicles within the global assessment, will

be registered in the database available for the control authorities for the compliance inspections of oversized and excessive transport use.

To create the global assessment of bridge passage in Slovakia, it is necessary to comply with the following marginal conditions:

10. SRA, Road Databank will register the bridges on the roads, fully requiring 100% of the data—at least on the bridging length, construction-technical condition, and load;
11. SRA, Road Databank will update the above-mentioned data over time on the basis of the data available;
12. SRA, Road Databank will register the data on all oversized and excessive transports with a total mass not exceeding 60 t and a total height and width not exceeding 4.5 m;
13. if there is a change in the construction-technical condition of the engineering structures of the road network (e.g., change in bridge load, or requirement of static assessment) that could affect the transit within given determinations, a relevant road transport administrator is obliged to inform SRA, Road Databank on that change;
14. if the global assessment will change due to the above-mentioned situation and, thus, the transport of vehicles/vehicle combinations will be restricted, SRA, Road Databank will immediately inform the haulier for who the global assessment was given. The haulier using the global assessment of bridge passage will be registered in IS RNM (Information System of Road Network Model) through which they will communicate with SRA, Road Databank;
15. if the global assessment does not require a static assessment of bridges relating to the transport, the permission from the administrators for particular road sections is not needed (there is a general permission for oversized and excessive transport given according to the global assessment results);
16. the haulier, if needed, requires a static assessment of selected bridges and the signature of these structures will be part of the permission for the special use of roads in relation to oversized and excessive transport.

## 4. Discussion

The global assessment of bridge passage introduces an entirely new approach within the procedure for obtaining a specific permit for road use as well as within transport realisation itself. In order to ensure the safety and continuity of road traffic and, based on the needs from bridge passage assessments due to the low presentation of abnormal axle load transport or transport with a large total weight, it is appropriate to define the limiting conditions under which the global assessment could also be applied into practice.

Therefore, the conditions of the global assessment for Slovakia are determined as follows:

17. the width and height of a vehicle/vehicle combination including the load do not exceed 4.5 m, and, concerning the vehicles used by armed forces or armed security forces, for which there is no requirement of special road use permit, the global assessment can be used when the total height of a vehicle/vehicle combination does not exceed 5 m;
18. the load of any axle does not exceed 14 t;
19. the number of the vehicle/vehicle combination's axles does not exceed 15;
20. the total mass of the vehicle/vehicle combination does not exceed 120 t;
21. all the requirements for the haulier transport capability under Technical Conditions TP 103 are enforced;
22. all the requirements for the escort of oversized and excessive transport under Technical Conditions TP 103 are enforced.

Due to missing data for CTC, bridging length, load rating of bridges, and the low frequency of updates, as well as the lack of data about transports up to 60 t and with a width and height up to 4.5 m, it is currently not possible to make a global assessment of all the bridges registered by SRA and for all the vehicles/vehicle combinations operated in Slovakia.

Since the legislation is amended constantly and the technical regulations are updated as well, it is also necessary to develop the applications and add other ways of passage, e.g., in normal transit mode for vehicle combinations that do not exceed the normal bridge load and in transit of vehicles/vehicle combinations within which the share of each vehicle combination's instantaneous mass and sum of the wheelbase are not larger than 2.7 t/m and their maximum permitted axle loads are not exceeded, pursuant to Section 5, par. 2 of Decree no. 134/2018 Coll. laying down details on road traffic vehicle operation, where normal load of bridge is to have a value of at least 32 t. The change in the exceptional transit mode occurred only on bridges designed according to Standard STN 73 6203/1986 for loading class A, which have a normal load of at least 32 t, an exclusive load of at least 80 t, and an exceptional load of at least 196 t. These bridges can be crossed, in an exceptional transit mode, by a vehicle combination with a total mass up to 280 t under the conditions that a load of any axles does not exceed 14 t, axle wheelbase is at least 1.4 m, distance between the last axle of the preceding vehicle and the first axle of the following vehicle is at least 4 m, and this vehicle/vehicle combination can cross the bridge only at night without accelerating or decelerating [55,56].

Similarly, the modified application can be used for determining the way for bridge passage abroad. For instance, in Austria, there is a document "Standard conditions for special transport, catalogue of conditions for bridge passage 2013" that defines the conditions for bridge passage in detail. As a rule, the bridges should be crossed at a constant speed without accelerating or decelerating. The bridge passage is not possible when there is congestion, narrowing due to accident, or presence of other special haulage, cane truck, or machine for bridge control. The extent of usage depends only on the number of conditions assessed as well as on the structure of the data on the bridges within certain areas [57].

The determination for a critical vehicle combination can be used in any country, but the global assessment of bridges must be in line with the national technical specifications for the load rating of bridges e.g., according to the different national annexes of EN 1991-2: Eurocode 1 and different national bridge transit modes.

Contributions of the global assessment of routes can be divided into three categories depending on the party affected:

23. contribution for SRA, Road Databank—possibility to make a global assessment for critical vehicles within selected intervals of mass;
24. contribution for the haulier—possibility to apply for a global assessment for selected vehicles, group of vehicles, whole fleet, or several numbers of wheelbases when using vehicles with changeable wheelbases;
25. contribution for armed forces and armed security forces—assessment for a critical vehicle or for selected fleet on a one-time basis without assessing each route individually.

Besides the above-mentioned contributions, the mobile application can be extended to other fields when using suitable software. The application could be a part of the control systems within road transport if the data provided for hauliers were sent to the control authorities as well. When interconnecting the application with GNSS equipment with the option of route recording, the observance of conditions given in the permission for special road use in relation to oversized and excessive transport could be controlled or used in the case of their breach (breach of determined time, departure from the route determined, or breach the speed determined for bridge passage). The application can also serve for monitoring vehicles in real time for their movements, which would mean that the sender, receiver, or other participant will be informed on the location of the cargo. When interconnecting the application with a system of dynamic weighing, it is possible to verify whether the total mass of a vehicle/vehicle combination is identical with the mass given in the permission for special road use. All these additional usages can be applied only if the global assessment of passage performed by using navigational applications will be classified as mandatory. The question is which authority shall perform the global assessment. In Slovakia, the data are processed by SRA, however, publicly available data on bridges are updated on an annual basis, and data of the conditions on 1 January is released

later, during the course of the second quarter of the year in question. The released data are often incomplete, causing problems in our assessment as well, and it was necessary to ask for one or more databases from road infrastructure administrators.

The research outputs support the implementation of information technologies, digitalization, and electronic communication into the process of oversized and excessive load transport, thereby aiming to preserve the road engineering constructions when breaching the procedures of the given permission for abnormal transport. In the future, it is possible to create the right conditions for online monitoring of all oversized and excessive transports in all the Member States in the EU and to contribute to the protection of road infrastructure, which is partially constructed through the EU Structural Funds, as well as be used in traffic management as a part of smart transport systems.

## 5. Conclusions

This article has paid attention to the possibilities of changing the authorisation procedure and realisation of oversized and excessive transport. Since it is necessary, in some cases for different reasons, to replace the vehicle/vehicle combination for which the route assessment is made, we have developed an application comparing the selected fleet from the perspective of the size of the cumulative wheelbases and their corresponding cumulative axle loads. Via this application, it is also possible to create a theoretical "critical vehicle" or "critical vehicle combination", respectively, which covers the parameters of all the vehicles/vehicle combinations assessed. Relating to assessment of a critical vehicle/vehicle combination, it is possible to select any vehicle/vehicle combination for excessive load transport from the fleet assessed.

Based on the analysis of the fleet, we have determined the possibility of a global assessment of bridge passage for vehicles/vehicle combinations with a maximum total mass of 120 t and a maximum total width and height of 4.5 m, which covers 1041 of 1859 transports assessed from 1 January 2016 to 31 December 2020—or 56%. For these transports, it is possible to draw up a list of the determined transit modes for the entire Slovak territory, which can be further displayed as a single map layer in the geographical information system, and these layers can be further exported into the application announcing the way of bridge passage needed for transport.

The model of the global assessment of bridges can be further developed to take into account the different classification criteria for bridge transit without restrictions, e.g., the masses of individual vehicles/vehicle combinations or increasing the normal load rating of bridges for normal traffic or exceptional transit mode. Further research can also be directed to the analysis of the transport time for vehicle combinations based on different bridge transit modes. Future research can also define new bridge transit modes, which can be used for the global assessment of bridges.

**Author Contributions:** Introduction J.G. and J.J.; literature review J.G., J.J., M.F., P.M. and. M.V.; materials and methods J.G., J.J., M.F. and M.V.; data curation J.G., J.J., P.M., M.F. and. M.V.; results J.G., J.J., M.F. and P.M.; writing—original draft M.F. and. J.J.; visualization J.J., M.F. and P.M. All authors have read and agreed to the published version of the manuscript.

**Funding:** This research was realized with support of Research project: P-101-0509/20, Technical Conditions TP 103—Excessive and oversized transport), funded by Slovak Road Administration, Bratislava, Slovakia.

**Institutional Review Board Statement:** Not applicable.

**Informed Consent Statement:** Not applicable.

**Data Availability Statement:** The data were provided by the Slovak Road Administration from 1859 assessed oversized and excessive shipments from the period from 1 January 2016 to 31 December 2020.

**Conflicts of Interest:** The authors declare no conflict of interest.

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
