# Peer review of "Global Assessment of Bridge Passage in Relation to Oversized and Excessive Transport: Case Study Intended for Slovakia"

_applsci, doi:10.3390/app12041931_

Round 1

Reviewer 1 Report

This is a very interesting and current topic, which is unfortunately quite neglected at the moment.
For this reason, I would like to appreciate the topicality of the topic.

Personally, I would welcome the more detailed statistical sample on which the research was based - but it is not necessary for a possible revision.

Author Response

Revision Cover Letter - Global assessment of bridge passage in relation to oversized and excessive transport: Case study intended for Slovakia

Jozef Gnap, Juraj Jagelčák, Peter Marienka , Marcel Frančák a Mária Vojteková

Reviewer 1

This is a very interesting and current topic, which is unfortunately quite neglected at the moment. For this reason, I would like to appreciate the topicality of the topic.

Personally, I would welcome the more detailed statistical sample on which the research was based - but it is not necessary for a possible revision.

Response 1: Thank you very much for your sincere opinion. There are two statistical samples included in the paper. One is described in lines 679-682 including figures 7, 8 and table 3. This statistical sample represents data about real assessed transports by Slovak Road Databank between 1.1.2016 – 31.12.2020 and represents 932 unique vehicles/combination vehicles and 1859 assessed transports. For the paper, we have used axle loads and wheelbases of this vehicles/combination vehicles. The second statistical sample represents parameters of bridges in Slovakia from Slovak Road Databank by different CTC. The second statistical sample is described in lines 737-749 and table 5.

Reviewer 2 Report

Dear authors, thank you for creating an interesting and thought-provoking case study on a very topical subject. The topic of your manuscript is very topical. The manuscript is of a very high standard. I recommend making some changes to the manuscript that would significantly improve its quality:

  • the statement in Abstract section (lines 25-27): "Thus, the authors try to know whether it is possible to design a global assessment of bridge passage in relation to heavy and oversize road transport while verifying it in the conditions for the EU Member State from Central Europe – Slovakia." would be appropriate to transform into the form of the aim of the manuscript without the use of "try to";
  • I would recommend splitting the keyword "oversized and excessive transport" (line 36) it into two separate keywords;
  • the literature search in section 1 is weak and it would be advisable to supplement it with other relevant sources in the field of the issue under study;
  • the work with references (lines 48-74) is not quite right, because references to two references are not always correctly listed after a paragraph and it is not clear which reference belongs to which statement;
  • between Chapter 2 and the first subchapter in Chapter 2 (2.1), there must be some introductory text (lines 313-314);
  • the Discussion section (4) lacks clearly stated limits of your manuscript and your case study.

I wish you every success in your further scientific research work.

Author Response

Revision Cover Letter - Global assessment of bridge passage in relation to oversized and excessive transport: Case study intended for Slovakia

Jozef Gnap, Juraj Jagelčák, Peter Marienka , Marcel Frančák a Mária Vojteková

Reviewer 2

Dear authors, thank you for creating an interesting and thought-provoking case study on a very topical subject. The topic of your manuscript is very topical. The manuscript is of a very high standard. I recommend making some changes to the manuscript that would significantly improve its quality:

Point 1

The statement in Abstract section (lines 25-27): "Thus, the authors try to know whether it is possible to design a global assessment of bridge passage in relation to heavy and oversize road transport while verifying it in the conditions for the EU Member State from Central Europe – Slovakia." would be appropriate to transform into the form of the aim of the manuscript without the use of "try to";

Response 1

Thank you very much for your sincere opinion. The sentence was changed as proposed.

Point 2

I would recommend splitting the keyword "oversized and excessive transport" (line 36) it into two separate keywords.

Response 2

The keywords were changed as proposed.

Point 3

The literature search in section 1 is weak and it would be advisable to supplement it with other relevant sources in the field of the issue under study.

Response 3

More references were added. (306-393)

Point 4

The work with references (lines 48-74) is not quite right, because references to two references are not always correctly listed after a paragraph and it is not clear which reference belongs to which statement.

Response 4

References 1-8 were corrected as proposed.

Point 5

Between Chapter 2 and the first subchapter in Chapter 2 (2.1), there must be some introductory text (lines 313-314).

Response 5

The introductory text was inserted as proposed.

Point 6

The Discussion section (4) lacks clearly stated limits of your manuscript and your case study.

Response 6

Global assessment is limited by the missing data about CTC of bridges, bridging length and load rating of bridges, low frequency of data updates, lack of data about transports up to 60 tonnes and with width and height up to 4,5 m. Clarification text was added to discussion. (859-862)

Reviewer 3 Report

Dear Authors,

Thank you for sending your paper to the journal Applied Sciences. Your paper deals with assessing bridges for transport of oversized and excessive shipments. The topic is interesting and in the scope of the journal.

I want to propose you some improvements to the current version of the paper:

  1. Try to leverage the use of words Transport (UK) and Transportation (US). So, your text needs to be written in the UK English language or US English language.
  2. Line 313 and 314 between titles of (sub)chapters need to have at least one paragraph of text.
  3. In subchapter 2.2, you start to explain your model. It is necessary to explain at the beginning what is the purpose of the modelling. What are input data and what is output data? Adding a block diagram can make it easy to understand your model.
  4. Line 438 – 441 is not clear. Where are you doing the calculation? Which tables are you using?
  5. Figure 1 – VC stand for? How you calculate VC?
  6. Line 451 – 452 is unclear: "There always three combinations assessed." Which three?
  7. Line 455 add 1
  8. Model examples are hard to read. I propose to you to add the figure of vehicles for better understanding.
  9. Line 521 – 528 can you show that in the process diagram?
  10. Adjust table 3 to be more readable. There is no need to point out the period of collecting the data. You have a point that in text.
  11. First, it will be better to explain current figure 6 and then introduce this in current table 5.
  12. Line 570 – is there any particular reason for picking up VC 25? Please explain.
  13. Line 573 – "data into application" – see my comment nr 4.
  14. In table 4, you used an axel until 24, but in Figure 7, you have 12 axels. Need to be clarified.
  15. Line 599 – 605 – you have a classification, and then in table 5, you have a new classification. For what purpose do you have those classifications? Is there any connection for the overall results of the assestment?
  16. You used different abbreviations (not only in this part of the paper), but you did not explain the meaning. All abbreviations need to be explained. For example: OSM and CTC.
  17. In Chapter Discussion, you have explained using your model only in Slovakia. Readers will be interesting at least how your model can be used on the level of the European Union. Ned to be commented.
  18. In chapter Conclusion, there is no statement of future research steps and research.

Regards,

Author Response

Revision Cover Letter - Global assessment of bridge passage in relation to oversized and excessive transport: Case study intended for Slovakia

Jozef Gnap, Juraj Jagelčák, Peter Marienka, Marcel Frančák a Mária Vojteková

Reviewer 3

Dear Authors,

Thank you for sending your paper to the journal Applied Sciences. Your paper deals with assessing bridges for transport of oversized and excessive shipments. The topic is interesting and in the scope of the journal.

Thank you very much for your sincere opinion.

Point 1

Try to leverage the use of words Transport (UK) and Transportation (US). So, your text needs to be written in the UK English language or US English language.

Response 1

The word transport was used in the whole text of paper except references.

Point 2

Line 313 and 314 between titles of (sub)chapters need to have at least one paragraph of text.

Response 2

The introductory text was inserted as proposed.

Point 3

In subchapter 2.2, you start to explain your model. It is necessary to explain at the beginning what is the purpose of the modelling. What are input data and what is output data? Adding a block diagram can make it easy to understand your model.

Response 3

The relevant sentences were added right behind 2.2 (458-464, 527-536) to explain input and output data of the model to understand the model better.

Point 4

Line 438 – 441 is not clear. Where are you doing the calculation? Which tables are you using?

Response 4

The sentences are now clearer (546-550).

Point 5

Figure 1 – VC stand for? How you calculate VC?

Response 5

The VC stands vehicle combination and now it is explained in Figure 1.

Point 6

Line 451 – 452 is unclear: "There always three combinations assessed." Which three?

Response 6

Clarification text was added (559-569).

Point 7

Line 455 add 1

Response 7

The number was added (574).

Point 8

Model examples are hard to read. I propose to you to add the figure of vehicles for better understanding.

Response 8

The general figure of semi-trailer vehicle combination was added as figure 2 with wheelbases li1-li5 and axle loads mi1-mi5 before Table 2. All VC’s in Table 2 are semi-trailer VC’s with different wheelbases and axle loads.

Point 9

Line 521 – 528 can you show that in the process diagram?

Response 9

The process diagram was added as Figure 6 and relevant text modified (627-660).

Point 10

Adjust table 3 to be more readable. There is no need to point out the period of collecting the data. You have a point that in text.

Response 10

The table 3 was changed as proposed.

Point 11

First, it will be better to explain current figure 6 and then introduce this in current table 5.

Response 11

The letters in Figure 8(old 6) were explained (700-701) under the figure and deleted from Table 3.

Point 12

Line 570 – is there any particular reason for picking up VC 25? Please explain.

Response 12

Explanation was added to the text (702-707).

Point 13

Line 573 – "data into application" – see my comment nr 4.

Response 13

The text was changed to be more clear (710-726).

Point 14

In table 4, you used an axel until 24, but in Figure 7, you have 12 axels. Need to be clarified.

Response 14

Clarification text was added to paper (729-735).

Point 15

Line 599 – 605 – you have a classification, and then in table 5, you have a new classification. For what purpose do you have those classifications? Is there any connection for the overall results of the assestment?

Response 15

Yes, the first classification is a results of assessment of each bridge and CTC is technical condition of the bridge. CTC is not used in  assessment of bridges but it is further used together with assessment results in figures 11,12,13. Current requirements in Slovakia describes special escort vehicles registered in Slovakia for CTC 6 bridges therefore the results were presented for CTC 1-5 (Figure 11) and CTC 6,7 (Figure 12). The text was also cleared in the paper (753-762).

Point 16

You used different abbreviations (not only in this part of the paper), but you did not explain the meaning. All abbreviations need to be explained. For example: OSM and CTC.

Response 16

All abbreviations are explained when occurring firstly in the text.

Point 17

In Chapter Discussion, you have explained using your model only in Slovakia. Readers will be interesting at least how your model can be used on the level of the European Union. Ned to be commented.

Response 17

The clarification sentence was added to discussion (887-890).

Point 18

In chapter Conclusion, there is no statement of future research steps and research.

Response 18

Future research steps and research was inserted at the end of conlusions (950-956).

Round 2

Reviewer 1 Report

Thanks for the explanation.

Reviewer 3 Report

Dear Authors,

Thank you for the new version of your paper. You have answered all my raised questions.

Regards,